# AdaFNIO: A Physics-Informed Adaptive Fourier Neural Interpolation Operator for Synthetic Frame Generation

## Abstract

We present, **AdaFNIO** - Adaptive Fourier Neural Interpolation Operator, a neural operator-based architecture to perform synthetic frame generation. Current deep learning-based methods rely on local convolutions for feature learning and suffer from not being scale-invariant, thus requiring training data to be augmented through random flipping and re-scaling. On the other hand, **AdaFNIO** leverages the principles of physics to learn the features in the frames, independent of input resolution, through token mixing and global convolution in the Fourier spectral domain by using Fast Fourier Transform (FFT). We show that **AdaFNIO** can produce visually smooth and accurate results. To evaluate the visual quality of our interpolated frames, we calculate the structural similarity index (SSIM) and Peak Signal to Noise Ratio (PSNR) between the generated frame and the ground truth frame. We provide the quantitative performance of our model on Vimeo-90K dataset, DAVIS, UCF101 and DISFA+ dataset. Lastly, we apply the model to in-the-wild videos such as photosynthesis, brain MRI recordings and red blood cell animations

## 1 Introduction

Video frame interpolation is an intricate process that generates multiple intermediate frames from a given set of available frames. This problem presents significant challenges due to the necessity of comprehending the geometric structures of images, predicting the positions of numerous objects within the images, and accounting for the complex velocities of these objects and the time steps between frames. In the context of biology and physics, this process can be compared to understanding the dynamic motion of biological systems, such as cellular structures or the interactions of particles in a fluid medium.

From the perspective of scientific computing, addressing the challenge of video frame interpolation requires the development of advanced algorithms and architectures that can efficiently handle the complex and dynamic nature of the problem. By incorporating insights from physics, it is possible to create more accurate and robust solutions for video frame interpolation that account for the intricacies of diverse systems and processes.

Moreover, an efficient video interpolation system must be compatible with commonly used devices, operate on edge hardware, and accommodate videos of any arbitrary resolution. Utilizing neural networks as a solution to interpolation offers a low-cost alternative, as devices only need to store the weights, which are typically a few hundred megabytes in size. Nevertheless, neural networks relying solely on convolutional filters face limitations in generalizing well to scaling. This is due to the fixed size of the filters, which can only recognize patterns that conform to their dimensions.

The challenges associated with video capture in various applications, such as wildlife monitoring Swann et al. (2011), remote sensing Campbell & Wynne (2011), microscopy Pawley (2006), underwater research Kocak et al. (2008), and space exploration Squyres et al. (2004), often necessitate the use of low frame rate or low-resolution videos. For instance, trail cameras placed in natural habitats are designed to conserve battery life and storage space, leading to lower frame rates and resolutions Swann et al. (2011). Similarly, remote sensing platforms, such as satellites or aerial vehicles, face data transmission limitations that can result in reduced video quality Campbell & Wynne (2011). In the field of microscopy, imaging hardware constraints

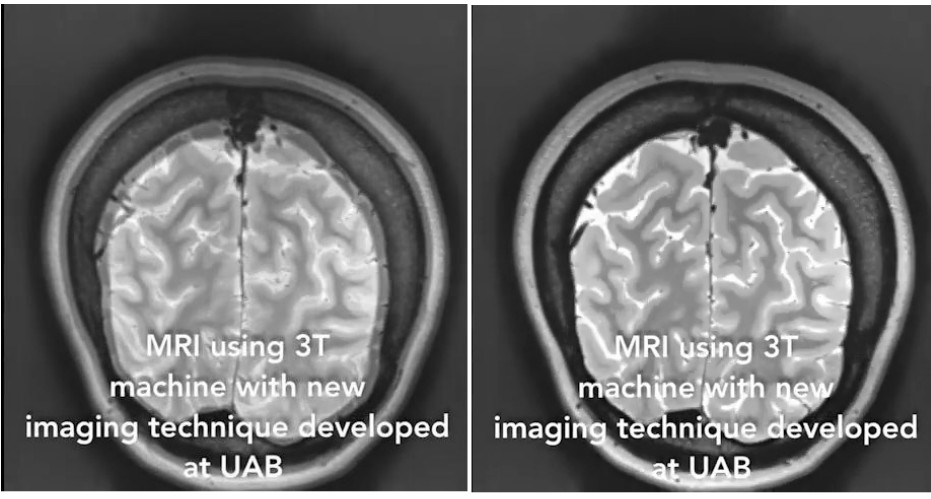

Figure 1: *The figure highlights the qualitative difference in the frames generated through the simple blending of adjacent frames versus synthetically generating them through the AdaFNIO model. It can be observed that the frame on the left is quite blurry, while the one on the right is clear and sharp*

and the need to minimize data generation during extended observation periods can lead to lower frame rates and resolutions Pawley (2006). Underwater research and exploration also demand video capture devices that can operate efficiently under challenging environmental conditions, such as low light levels or limited visibility, which may require lower frame rates and resolutions Kocak et al. (2008). Finally, space exploration probes and rovers often employ cameras that prioritize conserving power and storage resources, as well as accommodating limited bandwidth for data transmission back to Earth, resulting in lower frame rates and resolutions Squyres et al. (2004). A robust physics-based frame interpolation method that is also scale-invariant can greatly benefit these applications by enabling the accurate generation of intermediate frames, irrespective of the input video resolution. This would allow for more precise motion analysis, improved data visualization, and enhanced temporal resolution in low frame-rate or low-resolution videos. Additionally, such an interpolation technique would eliminate the need for extensive data preprocessing or resolution-specific model training, making it a versatile solution across different fields and video capture devices.

Many other applications make use of video frame interpolation, such as apps that generate movies and panoramic views from visually similar frames and applications that run on the network edge that may need to recover lost frames due to network issues, restore broken videos Cheng et al. (2021); Liang et al. (2022); Kim et al. (2018). Recent works focus on increasing the video frame rate to improve gaming and real-time video streaming experience. The other applications include medical imaging Ali et al. (2021); Karargyris & Bourbakis (2010), restoring compressed videos He et al. (2020) and generating virtual frames in 3D spaces Smolic et al. (2008); Wang et al. (2010). Most of these applications, especially video streaming applications and gaming environments, need the interpolation algorithm running on the network edge while also handling very high-resolution videos. Several cutting-edge models have been developed, and they produce interpolated frames that, on average, have a structural similarity of 98% with the expected output but are trained on small patches of input. While it is very expensive computationally to train neural networks on high-resolution inputs, it is possible to design architectures that can be trained on low-resolution inputs but generalize well to high-resolution ones. In this paper, we present a powerful neural operator-based architecture for interpolating video frames through token mixing that has a quasi-linear complexity and is independent of input resolution Guibas et al. (2021).

Moreover, an efficient video interpolation system must be compatible with commonly used devices, operate on edge hardware, and accommodate videos of any arbitrary resolution. Utilizing neural networks as a solution to interpolation offers a low-cost alternative, as devices only need to store the weights, which are

typically a few hundred megabytes in size. Nevertheless, neural networks relying solely on convolutional filters face limitations in generalizing well to scaling. This is due to the fixed size of the filters, which can only recognize patterns that conform to their dimensions. Current deep learning-based interpolation models are generally composed of Convolutional layers Niklaus et al. (2017); Niklaus & Liu (2018); Cheng & Chen (2020). These models construct the missing frames by extracting the structural information present in the input images using the appropriate filters or kernels. Convolutional layers exhibit shift invariance as they capture objects present in different regions of the image. However, they are not invariant to scale or rotation. To overcome this issue, images are randomly flipped and rotated to capture different orientations of the same object. Moreover, Convolution neural network-based models rely on local convolution for feature learning and require large amounts of training data and take a long time to converge. There have been attempts to solve this issue, for example, video interpolation using cyclic frame generation technique Liu et al. (2019), but it is not very accurate.

Optical flow-based techniques Krishnamurthy et al. (1999), which capture the motion of objects, have also been applied for frame interpolation. In this technique, the apparent motion of the pixels is captured and labeled as a flow vector. Using the estimated values of these flow vectors for each pixel, a missing frame can be generated. Flow-based techniques resolve the limitations imposed by inadequate kernel sizes in CNN-based methods and the frames can also be generated at a higher frequency per second, resulting in a smoother video. Optical flow-based methods fail while dealing with noisy frames due to the lack of necessary pixel information. It has also been observed that a combination of both kernel-based and optical flow-based methods Choi et al. (2020) with Deep learning techniques like transformers Shi et al. (2022) and GANs have low frame interpolation errors and also provide a great frame rate for a smooth video. However, GAN-based architectures suffer from modal collapse and thus cannot be generalized in an arbitrary fashion. They rely heavily on the distribution of input and would require re-training for a new distribution if the input space is changed. Transformer-based architectures have been shown to be very efficient Shi et al. (2022). However, due to their massively complex architecture, they require a lot of computing power and powerful GPUs to train. Long training times are an additional downside.

In this paper, we define the problem of video interpolation from a physics perspective. The problem can be defined as predicting the trajectories of objects, each moving with a different velocity in continuous space, similar to optical flow, but we present a way to capture the flow information efficiently using kernels. This problem is similar to predicting the trajectory of wind currents or ocean currents, for which neural operators Kovachki et al. (2021); Guibas et al. (2021); Rahman et al. (2022) have been shown to be exceptionally efficient, beating state-of-the-art weather prediction models Pathak et al. (2022). We present a powerful Fourier Neural Operator-based architecture - AdaFNIO - Adaptive Fourier Neural Interpolation Operator and show empirically that the model achieves state-of-the-art results. We quantify the quality of the results using the Peak Signal to Noise Ratio (PSNR) and the Structural Similarity Index and show that the SSIM of the generated frames is structurally similar to the ground truth.

The approach AdaFNIO network captures complex motions and overcomes degrees of freedom limitations imposed by typical convolutional neural networks and imposes resolution invariance to the AdaCoF network through spectral convolution layers. The network has a sequence of spectral convolution layers, which perform convolution upon translating the input to the Fourier space or spectral domain. The low-frequency regions are retained and the high-frequency ones are discarded because high-frequency points are specific to the particular input and may lead to overfitting, as shown in Li et al. (2020). The spectral convolution architecture of the AdaFNIO network is similar to an autoencoder network where the encoder layers contract the input space to capture the key information about the distribution of the input. The decoder expands it back to its original input space.

In this work, we have three main contributions:

- We present a powerful, efficient neural operator-based architecture to perform video frame interpolation whose performance is comparable to state-of-the-art interpolation models. To the best of our knowledge, AdaFNIO is the first to propose a resolution invariant architecture to solve this problem.

- We leverage the fact that learning in the Fourier domain allows for resolution-independent learning and allows for generalization to high-resolution images to capture finer details in high-resolution images that are harder to capture.

- We show that AdaFNIO can generalize well on the unseen data by testing it on DAVIS-90 Caelles et al. (2019), UCS101 Liu et al. (2017) and DISFA+ datasets. Mavadati et al. (2016).

- We also apply this model to various observations, animations, and simulations and show how our approach can provide a way to overcome hardware and resource limitations in recording scientific observations. Figure 1 provides an example of how the model can be used for medical imaging videos.

## 2 Background and Related Work

Existing video interpolation methods focus on creating techniques that can accurately capture an object's motion while accounting for occlusions and blurry frames. To deal with blurry frames, the pyramid module proposed in Shen et al. (2020) successfully restores the inputs and synthesizes the interpolated frame. Many other methods focus on estimating the bidirectional optical flow in the input frames Liu et al. (2020a); Niklaus & Liu (2020) and these methods are usually trained using deep learning models like neural networks Huang et al. (2022), generative adversarial networks Tran & Yang (2022), long short term memory (LSTM) Hu et al. (2021) and autoencoders Chen et al. (2017); **?**. However, these methods fail to generate smooth interpolation of the frames when dealing with large motions and occlusion in the input frames.

Bao et al. Bao et al. (2019) introduced depth aware flow projection layer that uses depth maps in combination with the optical flow to suppress the effect of occluded objects. In addition to bidirectional optical flow estimation, Niklaus & Liu (2018) uses pixel-wise contextual information to generate high-quality intermediate frames and also effectively handles occlusion. To capture non-linear motion among frames, a bidirectional pseudo-three-dimensional warping layer is used in Luo et al. (2022) that uses motion estimation and depth-related occlusion estimation to learn the interpolation. ST-MFNet Danier et al. (2022) uses 3D CNNs, and spatio-temporal GANs to estimate intermediate flows to capture large and complex motions.

The optical flow estimating models are accurate, but their calculations are expensive, and their designs are complicated. An alternate technique was kernel-based, Shi et al. (2021); Liu et al. (2019), which uses filters to learn features from the input frames in order to synthesize an intermediate frame. These models are end-to-end trainable but fail to capture motion and pixel information beyond the kernel size. To overcome these limitations and also to handle other major issues like occlusion Choi et al. Choi et al. (2020) proposed an architecture that uses a layer called PixelShuffle. The PixelShuffle layer in Choi et al. (2020) downsizes the input frames to capture relevant feature information and upscales the feature maps in later stages to generate the missing frame and is a replacement for flow-estimation networks. A similar model that uses transformers has been proposed in Kim et al. (2022), which uses a visual transformer module that analyzes input frame features to refine the interpolated prediction and also introduces an image attention module to deal with occlusion. To avoid the additional computation overheads of having an image attention module, Shi et al. (2022) uses transformers along with local attention, which is extended to a spatial-temporal domain to compute the interpolation.

To overcome the issue of restricted kernel size in kernel-based methods, Niklaus et al. (2017) introduces adaptive separable convolution that uses deep convolutional neural networks and runs pairs of 1D kernels on the input frames to capture large motions. In addition to kernel-based methods, there are phase-based methods like Meyer et al. (2015) that use per-pixel phase modification to compute a smooth interpolated frame without having to perform any flow estimation.

The above-discussed methods show exceptional results on well-known datasets like Vimeo-90K, but a major drawback is that these models take too long to converge. For example, DAIN Bao et al. (2019) takes 5+ days to converge on an NVIDIA Titan X (Pascal) GPU for around 40 epochs and a batch size of 2; CAIN Choi et al. (2020) is trained for 200 epochs with a batch size of 32 and takes around 4 days to converge on a single Titan Xp GPU.

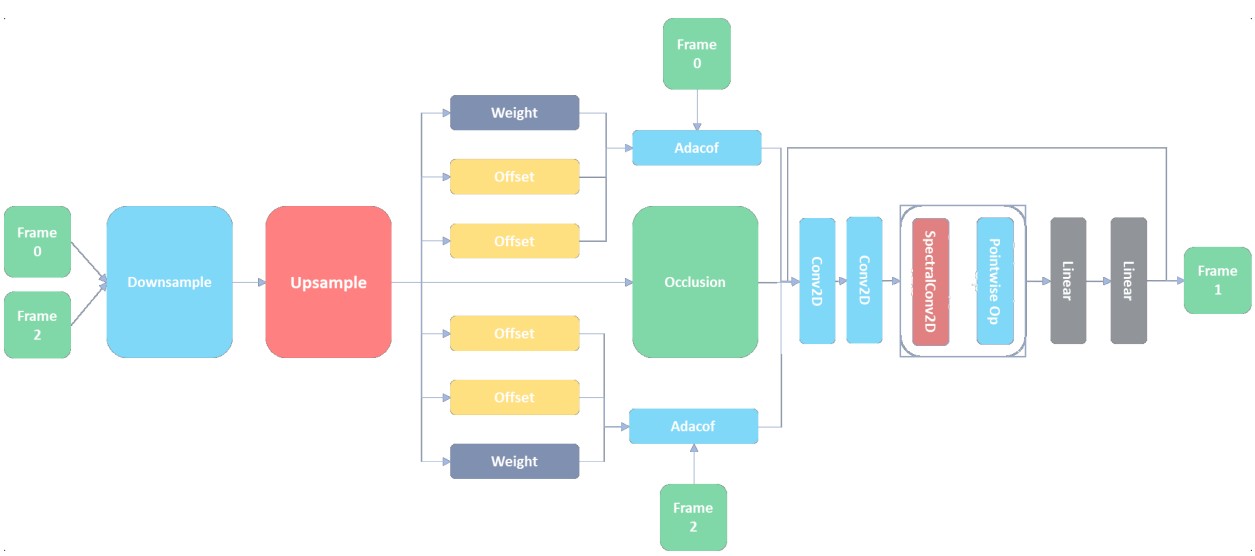

Figure 2: *Abstract view of the ADAFNIO Architecture: Our architecture combines the AdaCoF network with neural operator layers. Initially, the two frames are fed to the encoder (Downsample) network, which applies successive 2D Convolution to the frames to generate a latent representation. The Upsample block consists of a sequence of upsample and convolution layers to reconstruct the frame from the latent representation. This reconstructed matrix is fed to two sets of 3 subnetworks - 2 Offset networks and 1 Weight network. These sub-networks extract the features for the AdaCof layer, which processes these features with the input frames. The final output is fed to the neural operator layers to extract finer information not captured by AdaCoF. The SpectralConv layers perform pixel-wise multiplication in the Fourier domain by first performing FFT on the frames. This process is analogous to token mixing. The final output is a weighted sum of the AdaCoF output and the Neural Operator output*

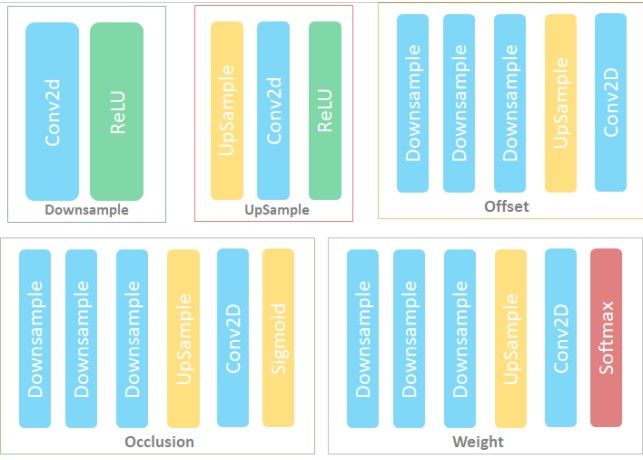

Figure 3: *The figure shows the constituent layers of each block in the AdaCoF sub-network*

## 3 AdaFNIO Architecture

AdaFNIO aims to overcome one of the common limitations exhibited by models that use convolutional layers - a variance to scale and makes the model resolution independent. Adaptive collaboration of flows, or AdaCoF, is a state-of-the-art architecture that improves the degrees of freedom of complex motions in the frames. The AdaCoF architecture offers a very generalized solution to determining the flows, irrespective of the number of pixels and their location in the frame Lee et al. (2020). To generalize AdaCoF to any

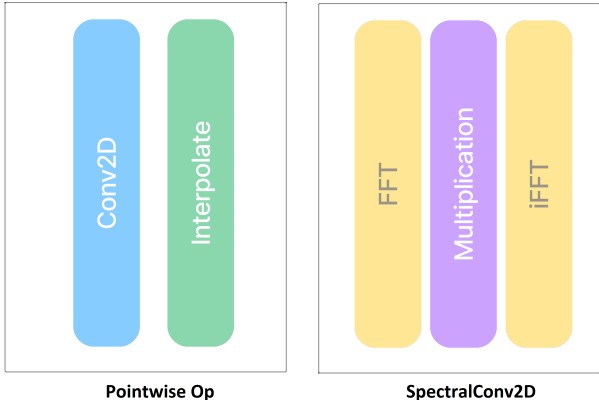

Figure 4: *The figure shows the constituent operations performed in each of the blocks of the Neural Operator sub-network*

arbitrary resolution, we propose connecting neural operator layers to the model to capture finer information that is otherwise hard to generalize at higher resolutions. The final output is a weighted sum of the features learned by the neural operator and the flows learned by the base AdaCoF model.

The proposed model AdaFNIO takes as input a couplet of frames and generates the intermediate frame. As shown in equation 1, $I_0$ and $I_1$ are the input frames and $I_{0.5}$ is the interpolated frame. $w_1$ and $w_2$ are weights chosen for the features generated by the two models, which are tuned during training. If $I'_{0.5}$ is the ground truth and $\mathcal{N}$ is the neural operator network, then the resulting frame generation process is described as follows

$$I_{0.5} = w_1 \mathcal{N}(I_0, I_1) + w_2 AdaCoF(I_0, I_1) \tag{1}$$

The interpolated frame quality is measured as PSNR( $I'_{0.5}$, $I_{0.5}$ ) and SSIM( $I'_{0.5}$, $I_{0.5}$ ).

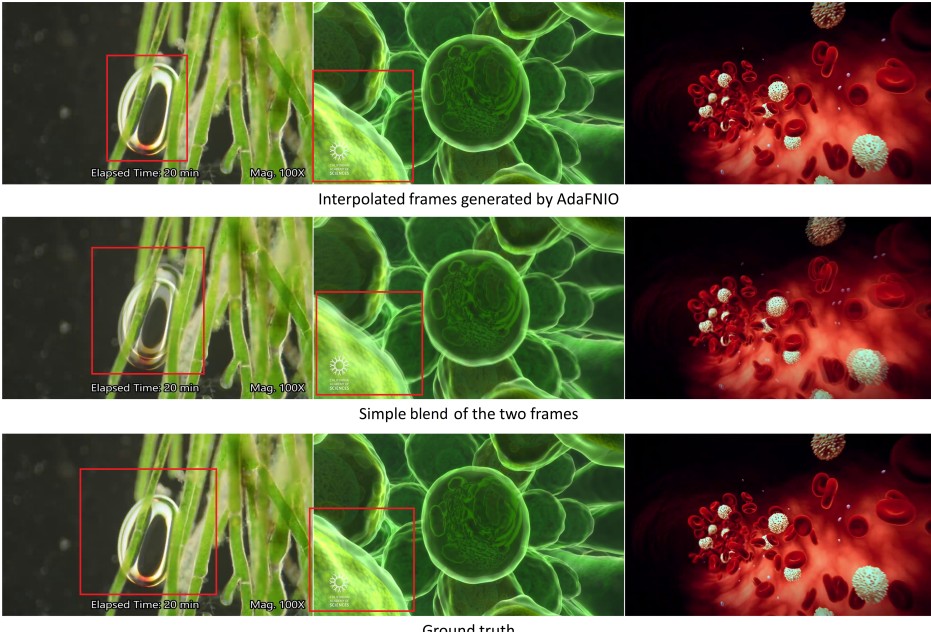

Figure 5: *This figure is a visual comparison of results generated by AdaFNIO against various video recordings and simulated videos. The leftmost column is a real-time recording of photosynthesis. The middle frame is an animation of plant cells, while the rightmost one is a simulation of red blood cells within the body*

### 3.1 Frames as a continuous set of tokens

The input to the model is a set of frames, which are images. Neural operators have been traditionally applied to solve partial differential equations or PDEs where the input is the discretization of a continuous vector field. Images, on the other hand, have distinct objects, sharp edges, and discontinuities. Recently, neural operators have shown promise in vision problems Guibas et al. (2021). In the case of frame interpolation, the frames can be broken down into a set of tokens, where each token is a patch within the frame. The model performs global convolution as a token mixing operation in the Fourier space. The tokens are extracted in the initial convolution layers with shared weights. The kernel size and the stride length determine the dimensions of the tokens. To account for non-periodicity in images, a local convolution layer is added after each global convolution.

### 3.2 Resolution Invariance and Quasi-linear Complexity

The token features are learned in the Fourier space, which is invariant to the resolution of the input image. This was shown in Guibas et al. (2021). This allows the model to exhibit the property of zero-shot super-resolution; that is, the model can be trained on one resolution and tested on any arbitrary resolution.

The multiplication is done on the lower $(k_x, k_y)$ Fourier modes, which is restricted to be at most $(n/2, m/2)$, where $(n, m)$ is the resolution of the image. If the weight matrix has a dimension of (p, p), then the time complexity of global convolution is $O(Nlog(N)p^2)$, where N is the length of the token sequence, as shown in Guibas et al. (2021).

The model contains two components - The interpolation layer, which performs a linear operation on the two frames and couples them together with a common weight matrix. The second component is a version of the UNO network, similar to what was proposed in Rahman et al. Rahman et al. (2022), which is a neural operator network. It performs convolution operations in the Fourier Space. The UNO part of the model, termed NIO, initially contracts the input space to extract the key features of the image, which is equivalent to performing a dimensionality reduction. The encoder is followed by the decoder layers, which expand space back to its original size. The input takes two channels, with each channel corresponding to an input frame, while the output has a single channel corresponding to the output frame.

### 3.3 AdaFNIO Architecture

The AdaFNIO model comprises of two sub-networks as shown in figure 2. The AdaCoF network comprises of a U-Net implemented with convolutional layers and 6 networks that calculate 2 pairs of weight and offset vectors for the two frames. The individual blocks used in these networks are shown in figure 3.

The neural operator network initially extracts the tokens through a sequence of convolution layers with shared weights. This means that the two frames are fed to the convolution layers recursively. This forms the input to the spectral convolution layers that perform global convolution in Fourier space. The pointwise operation layers perform local convolution and resize the frames to the required dimensions. The output features of each layer are a weighted sum of these two outputs. Through a sequence of spectral convolution and pointwise operator layers, the input tensor is downsampled and the latent embedding is generated. This embedding is used to generate the output frame through another sequence of spectral convolution and pointwise operator layers that upsample the input tensor successively. The constituent components of the spectral convolution block and pointwise operator block are shown in figure 4.

The spectral convolution layer only preserves the low-frequency Fourier models and ignores the high-frequency modes, which are too specific to the particular input and if these modes are learned, they overfit to the input as shown in Li et al.Li et al. (2020). After applying the weights, the tensors are projected back into the spatial domain and non-linear activation is applied to recover the high-frequency points.

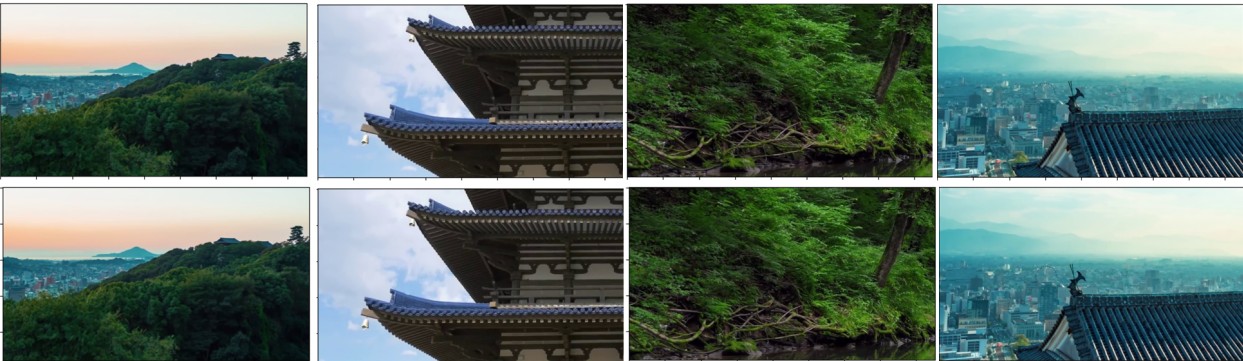

Figure 6: This figure is a visual comparison of results generated by AdaFNIO against high-resolution (1080p) stock footage of the Japanese landscape. While the visual differences are hard to discern, the AdaFNIO has a slightly better quantitative performance against AdaCoF. The top row is the output generated by AdaFNIO and the bottom row is the ground truth.

### 3.3.1 Loss Functions

The model uses two loss functions. The L1 loss function is used to initially train the model. This loss is given by equation 2

$$L_1 = ||I_{AdaFNIO} - I_{GT}||_1 \qquad (2)$$

The other loss function that is used is perceptual loss, which is used for fine-tuning the model. The loss is generated by the feature extractor of the pre-trained VGG22 neural network. This loss is given by equation 3

$$L_{vgg} = ||F(I_{AdaFNIO}) - F(I_{GT})|| \qquad (3)$$

The overall loss function used during fine-tuning is a combination of L1 and VGG loss functions, with higher weights given to the L1 loss function. This is denoted by equation 4

$$L = L_1 + 0.01 * L_{vgg} \qquad (4)$$

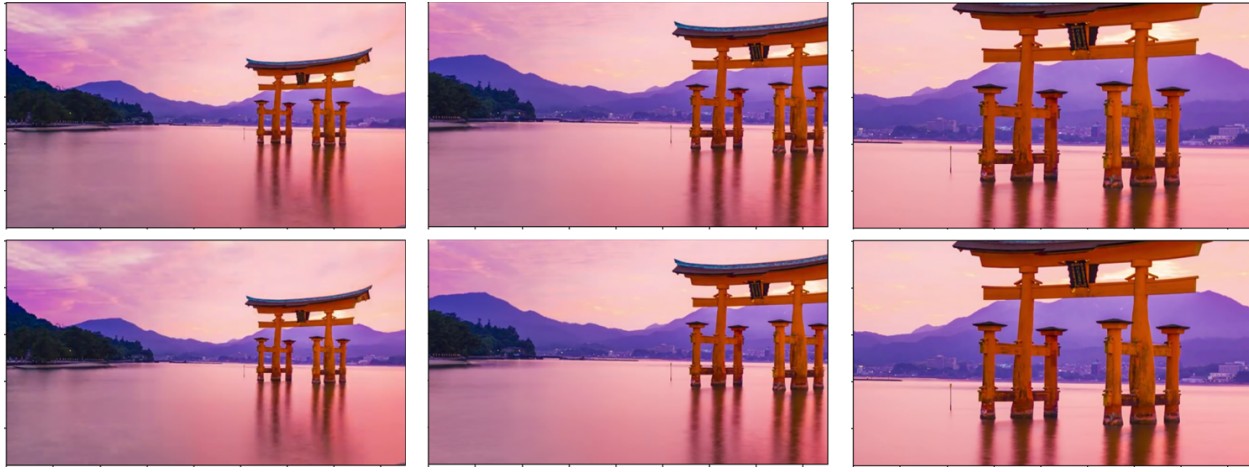

Figure 7: *The above figure highlights the resolution invariance property exhibited by the architecture. The models were trained on 256x256 patches of the Vimeo90K dataset but tested against high-resolution stock footage of the Japanese landscape. In this figure, the top row is the generated output and the bottom row is the ground truth. The left row is of 480p resolution, the center one 720p and the right one 1080p*

Table 1: *The table showing the quantitative performance on Vimeo90K, DAVIS, UCF101 and DISFA+ dataset. The AdaFNIO model is compared against the quantitative performance of other models, as presented in Kong et al. Kong et al. (2022) and Shi et al. Shi et al. (2022)*

| Model | Parameters (M) | Epochs | *Vimeo90K* | | *DAVIS* | | *UCF101* | | *DISFA+* | |
|---|---|---|---|---|---|---|---|---|---|---|
| | | | PSNR | SSIM | PSNR | SSIM | PSNR | SSIM | PSNR | SSIM |
| ToFlow Xue et al. (2019) | 1.4 | - | 33.73 | 0.968 | - | - | 34.58 | 0.966 | - | - |
| IFRNET-S Kong et al. (2022) | 2.8 | 300 | 35.59 | 0.978 | - | - | 35.28 | 0.969 | 38.85 | 0.961 |
| VFIT-S Shi et al. (2022) | 7.5 | 100 | 36.48 | 0.976 | **27.92** | 0.885 | **33.36** | **0.971** | 39.25 | 0.964 |
| SoftSplat Niklaus & Liu (2020) | 7.7 | - | 35.76 | 0.972 | 27.42 | 0.878 | 35.39 | 0.952 | 38.33 | 0.954 |
| RIFE Huang et al. (2020) | 9.8 | 25 | 35.62 | 0.978 | - | - | 35.28 | 0.969 | 38.84 | 0.961 |
| BMBC Park et al. (2020) | 11.0 | - | 34.76 | 0.965 | 26.42 | 0.868 | 35.15 | 0.968 | - | - |
| ABME Park et al. (2021) | 18.1 | - | 36.18 | **0.980** | - | - | 35.38 | 0.969 | - | - |
| SepConv Zhang et al. (2018) | 21.6 | - | 33.60 | 0.944 | 26.21 | 0.857 | 34.78 | 0.966 | 38.70 | 0.959 |
| AdaCof Lee et al. (2020) | 21.8 | 50 | 34.47 | 0.973 | 26.49 | 0.866 | 34.90 | 0.968 | 38.98 | 0.961 |
| DAIN Bao et al. (2019) | 24.0 | 40 | 33.35 | 0.945 | 26.12 | 0.870 | 34.99 | 0.968 | 35.00 | 0.956 |
| QVI Liu et al. (2020b) | 29.2 | 200 | 35.15 | 0.971 | 27.17 | 0.874 | 32.89 | 0.970 | - | - |
| SuperSloMo Jiang et al. (2018) | 39.6 | 500 | 32.90 | 0.957 | 25.65 | 0.857 | 32.33 | 0.960 | - | - |
| FLAVR Kalluri et al. (2020) | 42.4 | 200 | 36.30 | 0.975 | 27.44 | 0.874 | 33.33 | 0.971 | - | - |
| CAIN Choi et al. (2020) | 42.8 | 200 | 34.76 | 0.970 | 27.21 | 0.873 | 34.91 | 0.969 | - | - |
| **AdaFNIO** | 88.9 | 100 | **36.50** | 0.976 | 27.90 | **0.888** | 34.88 | 0.970 | **39.30** | **0.965** |

### 3.4 Frame Generation

The intermediate frame is generated after applying a sequence of spectral convolution layers that perform global convolution in Fourier space. Let the input frames be represented as $I_i$ and $I_j$ and the initial weight matrix be represented as $V_f$. Let the initial convolution layers be represented with $C_t$. Let the downsampler layer be represented as $E$ and the upsampler layer be represented as $D$. Let the weights be represented as $W$ and bias be represented with $B$ and AdaCoF network be represented by $Ada$. The pipeline is as follows

$$C_t(I_i, I_j) = V_f * I_i + V_f * I_j + B \tag{5}$$

$$I_{0.5} = W_{NIO} * D(E(C_t(I_i, I_j))) + Ada(I_i, I_j) \tag{6}$$

### 3.5 Training

For the training process, Vimeo90K triplet dataset was used. The dataset has 73,171 3-frame sequences, of which 58,536 frames were used for training and the remaining 14635 were used for validation. The L1 loss was used for the first 80 epochs. For the finetuning process, 11,000 random frames were used to tune the model for another 20 epochs with perceptual loss. The model was tested against 1080p stock footage of a Japanese landscape taken from YouTube channel 8K World. This video was chosen because the footage was shot in very high resolution. The model was trained for 100 epochs on Nvidia A100 GPU. The frames were randomly cropped to 256x256 patches. However, the frames were not randomly flipped, scaled, or rotated in order to test the invariance properties of the architecture.

#### 3.5.1 Dataset

The AdaFNIO model was validated on these three frequently used benchmark datasets (Vimeo90K, DAVIS, UCF101) as well as one specialty dataset (DISFA+), which focuses on videos with human faces.

- **Vimeo90K Dataset** Xue et al. (2019) The Vimeo90K dataset is built from 89,800 clips taken from the video streaming site Vimeo. It contains a large variety of scenes. For this project, the triplet dataset was used. The dataset contains 73,171 3-frame sequences, all of which have a resolution of 448x256.

- **Davis-90** The modified Densely Annotated Video Segmentation dataset contains frames from various scenes. These frames are partitioned into triplet sets and used for testing the performance of the model.

- **UCF101 Dataset** The preprocessed UCF101 dataset is a collection of scenes that have been partitioned into triplets. This dataset is also used for testing the models.

- **DISFA+ Dataset** The DISFA+ or the Denver Intensity of Spontaneous Facial Action Database consists of a large set of facial expression sequences, both posed and non-posed. The dataset has multiple subjects of different ethnicities exhibiting various facial expressions and is a comprehensive dataset to study micro facial expressions. This dataset was chosen due to the increase in the prevalence of video meetings and social media videos, many of which predominantly features human faces.

The DISFA+ dataset was processed into triplets and the model was trained to predict the second frame from the first and third frames. The Vimeo90K dataset was used to provide a comparison benchmark against other deep learning-based interpolation approaches, while the DISFA+ dataset was used to predict facial expressions from up close. This served as a test to determine the ability of neural operator-based models to interpolate minute facial muscle movements. The frames were resized to 256x256 due to memory and GPU constraints.

### 3.6 Hyperparameters

The models were built using Pytorch and trained on Nvidia A100 GPUs. The Fourier modes used for the layers are 5, 10, 21 and 42. The batch size was set to 32 and the learning rate was set to 0.0001. The weight for the NIO base model was set to 0.01. The training was done using Cuda 11.0. Adam optimizer was used with a weight decay of 0.0001, $\beta_1$ of 0.9 and $\beta_2$ of 0.999. The loss function used for training was a mean squared error (MSE) or L2 Loss.

## 4 Experiments and Evaluation

### 4.0.1 Quantitative performance

In this section, we provide a comparison benchmark against the other state-of-the-art models with SSIM and PSNR as the evaluation metrics. Table 1 shows the quantitative performance against other state-of-the-art models. AdaFNIO has the best PSNR (36.50) on the Vimeo90K dataset, and the best SSIM (0.888) on the DAVIS dataset and outperforms every other model against DISFA+ dataset. The reason for better performance against the DISFA+ dataset is that neural operator-based models perform well on periodic images with smooth edges. Talking head videos have the fewest number of objects and fewer edges within the frames and thus, neural operators outperform in that situation. This phenomenon was first identified by Guibas et al. (2021) and empirically verified by us.

Figure 7 shows the outputs generated by the model at three different resolutions 480p, 720p and 1080p, respectively, and there is no performance degradation as the resolution increases. Figure 9 highlights the performance of AdaFNIO against other state-of-the-art models.

### 4.1 Resolution Invariance and Scale Invariance

To showcase AdaFNIO's resolution and scale invariance, we perform inference on un-seen videos of previously unseen resolutions. In Table 3, we highlight the model's performance at higher resolution samples of the same dataset (YouTube footage video captured at 60FPS, 720p, 1080p, 2160p, and 4320p) to evaluate resolution invariance. We also show more comparative results in Table 2 and Figure 8. The kernel sizes used in convolution models are fixed (3x3, 5x5, etc.) This allows for learning features within the kernel effectively. If the object's size changes (i.e., higher resolution), the network will not learn. However, the Fourier domain focuses on frequency changes within the image and learns these parameters; therefore, using Fourier Neural Operators on tokenized images is more effective.

Table 2: *Comparisons of interpolation models at different resolutions, showing scale-invariance property of AdaFNIO*

| Model | 720p | 1080p | 2160p | 4320p |
|---|---|---|---|---|
| **AdaFNIO** | 0.9879 | 0.9920 | 0.9742 | 0.9393 |
| AdaCof | 0.9879 | 0.9919 | 0.9591 | 0.9113 |
| VFIT-S | 0.9764 | 0.9770 | 0.9116 | 0.8839 |
| FLAVR | 0.9653 | 0.9840 | 0.9210 | 0.8552 |
| CAIN | 0.9834 | 0.9582 | 0.9123 | 0.8721 |

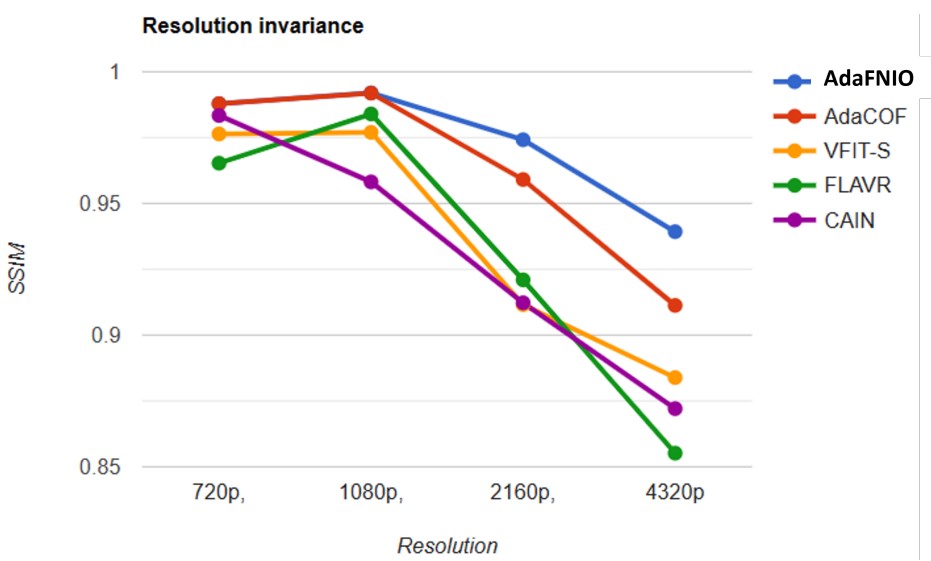

Figure 8: *Trends in performance at different resolutions*

## 4.2 Comparisons against baseline AdaCoF model

In this section, we show quantitative differences between the frames generated by the AdaCoF model and the AdaFNIO model in two settings - varying resolutions and varying frame rates. The models were tested against the Japanese stock footage video at 30fps, with 9,894 frames.

**Varying resolutions** At lower resolution, we observed that AdaFNIO and AdaCoF had similar performance, but as the resolution increased, AdaFNIO performed slightly better than the AdaCoF model. These SSIM values at different resolutions are shown in table 3

Table 3: *The SSIM values against Japanese stock footage video captured at different resolutions*

| Model | 480p | 720p | 1080p |
|---|---|---|---|
| AdaFNIO | 98.276 | 98.792 | 99.207 |
| AdaCoF | 98.276 | 98.790 | 99.199 |

**Varying frame rates and missing frames** To test the performance of the model with missing frames, the model was evaluated in three settings against 480p resolution frames - when every alternate frame was dropped, when 2 consecutive frames were dropped and when 4 consecutive frames were dropped. AdaFNIO slightly outperformed AdaCoF in all three settings and the SSIM values are shown in table 4

## 4.3 Ablation Study

As part of the ablation study, we tested different neural operator models against variants of Vimeo90K dataset in different settings. We refer to pure neural operator models, that is, models that were not combined with

Table 4: *The SSIM values against Japanese stock footage video captured at a fixed resolution of 480p but with varying frame rates*

| Model | drop 1 | drop 2 | drop 4 |
|---|---|---|---|
| AdaFNIO | 98.257 | 94.909 | 89.988 |
| AdaCoF | 98.256 | 94.899 | 89.969 |

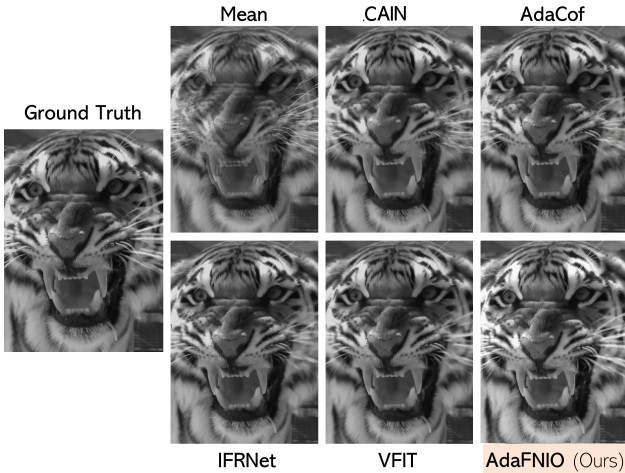

Figure 9: *This figure provides a visual comparison of results across various models. NIO achieves visually comparable results with other models*

AdaCoF, as NIO. These models are highly sensitive to sharp edges and generally perform well on low-resolution frames due to them having relatively smooth boundaries and fewer sharp objects. To remedy this issue, the neural operator was combined with AdaCoF in the final version of AdaFNIO. We shall discuss our findings on pure neural operator models in the following subsections.

We built two models - a basic single-channel NIO-base model with a single neural operator block with 3.42 million parameters and a lighter model - NIO-Small (NIO-S), with 1.18 million parameters and trained the model on the grayscale Vimeo-90K dataset. The reason for choosing a single-channel neural network was to reduce the distraction created by RGB channels while learning to interpolate. The model was trained to recognize the trajectories of the objects within the frame as the initial step of learning. The NIO-S model was not as accurate as the NIO-base model, but it was faster and took half as much time per epoch. On the A30 GPU, the NIO-base model took 5 and a half minutes per epoch, while the NIO-S model took 3 minutes and 39 seconds per epoch on average. This model only had three spectral convolution layers. The two models approached an SSIM of 0.90 within the first epoch. Table 5 highlights the differences between a single NIO block but with a different number of spectral layers when tested against the low resolution (50x50) DISFA+ dataset.

**Neural Operators and low-resolution RGB images** Three models were tested against a low-resolution RGB version of Vimeo90K dataset at 2 different resolutions of 100x100 and 85x85. These models were all trained for 50 epochs at a batch size of 32. The first model was the NIO fine model with 8 neural operator blocks connected to form a residual network, the second was an NIO base model with 4 neural operator blocks, and lastly, the AFNO Net with channel attention, tested against two patch sizes of 1x1 and 2x2. The models were tested against both normalized and non-normalized datasets. The SSIM values after convergence are shown in table 6.

It can be observed from table 6 that the AFNO network, which uses channel attention, performed poorly in every setting. NIO-based models that used downsamplers and upsamplers were quick to converge, as seen

Table 5: *The table showing the differences between NIO-S and NIO models*

| Model | Spectral Layers | Time/Epoch fixed batch | DISFA+ PSNR | SSIM |
|---|---|---|---|---|
| NIO-S | 4 | 00:03:39 | 39.25 | 0.987 |
| NIO | 8 | 00:05:43 | 38.84 | 0.954 |

Table 6: *The SSIM values low-resolution RGB Vimeo90K dataset for pure neural operator models. The N refers to the normalized dataset*

| Model | 100x100 | 100x100 N | 85x85 | 85x85 N |
|---|---|---|---|---|
| NIO base | 84.526 | 86.438 | 90.235 | 91.876 |
| NIO fine | 88.781 | 89.623 | 92.693 | 93.451 |
| AFNO 1x1 | 74.376 | 74.298 | 76.851 | 76.394 |
| AFNO 2x2 | 77.368 | 77.925 | 79.422 | 79.631 |

in table 5, and performed better than AFNOs. Therefore, the NIO-based models were used as the neural operator component in the final AdaFNIO model.

### 4.3.1 Normalization

Min-Max normalization was applied to the dataset as a preprocessing step, reducing the range of values that a pixel can take. The models performed slightly worse on the Vimeo90K dataset when the input was normalized. However, these models performed better on the other datasets after being trained on the normalized grayscale 100x100 Vimeo90K dataset, thus showing that normalization as a preprocessing step helps the models generalize better on data they have not seen before.

Table 7: *The table showing the differences between Normalizing the input and not normalizing the input*

| Model | UCF101 PSNR | SSIM |
|---|---|---|
| NIO | 36.54 | 0.970 |
| NIO + Norm | 36.84 | 0.954 |

## 5  Discussions and Future Use

Apart from the video datasets such as DISFA+, Vimeo90K and Davis, AdaFNIO was specifically tested on in-the-wild videos from various domains. These videos were a mixture of microscopic recordings, MRI recordings, scientific animations and simulations. Figure 5 depicts a visual comparison of outputs generated by the AdaFNIO model against a simple mean of the adjacent frames. It is clearly apparent that the model estimates the trajectories and places the objects in the right points within the frames, while a simple mean of the frames leads to a messy overlap of the objects. The figure depicts the synthetic frames generated from the following three videos -

- Microscopic footage of oxygen bubbles generated during photosynthesis of the aquatic plant anacharis (Egeria densa) and a Marimo ball (Aegagropila linnaei), uploaded by Sci-Inspi.

- An animation of Redwood leaf, highlighting stoma, palisade cells, golgi apparatus, endoplasmic reticulum, and ribosomes, uploaded by California Academy of Sciences.

- Blood component animation video highlighting red blood cells, white blood cells, plasma, and platelets, uploaded by American Society of Hematology.

Furthermore, the model was also used to interpolate MRI brain scans, uploaded by the University of Alabama, Birmingham.

While AdaFNIO is primarily designed for computer vision, video enhancement, and video processing tasks, the underlying technology will find applications in other domains, including:

- **Scientific simulations:** In fields like fluid dynamics or meteorology, adaptive frame interpolation could help improve the accuracy and efficiency of simulations by filling in gaps between time steps or refining existing models Janiga & Thévenin (2013); Park & Xu (2013).

- **Robotics and control systems:** AdaFNIO's ability to predict intermediate states could be useful in designing more efficient and smooth control algorithms for robots and other automated systems Kober et al. (2013).

# 6 Conclusion

In this paper, we have presented a neural operator-based architecture for performing frame interpolation. The model is powerful, resolution invariant, and discretization invariant, and achieves state-of-the-art performance on unseen datasets. The model has proven effective in capturing information in tiny regions of the image (tokens) and generalizing well in larger images. The AdaFNIO model has only been trained on a triplet dataset with consecutive frames. However, it remains to be seen how well it performs when trained against larger sequences of frames. Secondly, the resolution invariance is an important property of the neural operator, and it remains to be seen whether this can be used to improve the resolution of the images.

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

# A    Appendix

You may include other additional sections here.

