# OpenReview forum: "AdaFNIO: A Physics-Informed Adaptive Fourier Neural In- terpolation Operator for Synthetic Frame Generation"
_TMLR — Withdrawn by Authors_

### Review · Reviewer_xvsJ · 2024-05-18

**Summary Of Contributions:**

This paper introduces AdaFNIO, a new method for video frame interpolation that combines AdaCoF, a convolutional network that adapts to complex motions, and neural operator, a resolution-invariant and physics-informed approach that learns in the Fourier domain. The paper provides a thorough evaluation of the proposed method on four benchmark datasets (Vimeo90K, DAVIS, UCF101, and DISFA+) and achieves the SOTA results on most tasks. Furthermore, due to the Fourier layer adopted in AdaFNIO, the proposed method enjoys the resolution invariance property.

**Audience:**

Yes

**Broader Impact Concerns:**

I think the researchers in the CV field would be interested in this paper and related topics. Moreover, this paper does not have ethical risks.

**Claims And Evidence:**

No

**Requested Changes:**

I think a major revision is required before the manuscript is accepted. The major concerns are listed as follows:

1.	The authors choose 0.01 as their hyper-parameter in Eq.4. How did the authors choose this hyper-parameter?
2.	What is the training and inference time of the proposed method and baseline methods? It would be better to report them in Table 1.
3.	According to Table 1, the size of AdaFNIO is significantly larger than other baseline methods. Why not align the number of parameters?
4.	Compared to the baseline VFIT-S, VFIT-S obtains better results on many tasks, as shown in Table 1. I suggest comparing VFIT-S and AdaFNIO in more detail in order to highlight the advantages of the proposed methods.
5.	The **Resolution Invariance** is one of the most important contributions in this paper, while only one dataset is conducted in Sec. 4.1. It would be better to verify the claim on more datasets.
6.	The writing of this paper should be further improved, such as:

- In Sec. 1, the author claims *three* main contributions but lists four points.
- There are many different kinds of AdaCoF in this paper. For instance, we can see AdaCof and AdaCoF in the caption of Figure 2, and Adacof in Figure 2.
- (p, p) in Sec. 3.2 should be $(p, p)$.
- What is $F$ in Eq. 4?
7.	Some citations in this paper seem to be confused, such as:

- Bao et al. Bao et al. (2019) introduced … in Sec. 2.
- Missing citation in Sec. 2. (autoencoders Chen et al. (2017); ?.)
- Some papers’ citations should be updated, such as FNO (published in ICLR 2021) and U-NO (published in TMLR).

Minor issues may need to be clarified:

1.	It would be better to clarify the dataset in the caption of the figures and tables, such as Table 2 and Figure 9.

2.	It would be better to show more examples like Fig. 9 in the Appendix.

3.	The Fourier neural operator was first applied to the PDE-solving tasks. The advantages of AdaFNIO might be more obvious when conducting the experiments on the Navier-Stokes equation simulation tasks compared with other baseline methods.

**Strengths And Weaknesses:**

Strengths:
- The paper claims that AdaFNIO is the first work to use neural operators for video interpolation tasks and can achieve zero-shot super-resolution.
- The paper provides a thorough evaluation of the proposed method on various benchmark datasets (Vimeo90K, DAVIS, UCF101, and DISFA+) and a comparison with dozens of methods.

Weaknesses:
- The primary concern for me is that the novelty of this paper seems to be limited. The proposed method appears to be a simple combination of AdaCoF and FNO, and I am not sure whether such a contribution is enough for publication in TMLR.
- The experiments conducted in this paper can be further improved.
- The writing and presentation of this paper should be further improved before it can be accepted.

---

### Review · Reviewer_Rw97 · 2024-06-19

**Summary Of Contributions:**

This paper introduces a  neural architecture called AdaFNIO, which is designed for video frame interpolation.  It leverages the advantages in the Fourier domain to achieve resolution-independent learning. This approach allows the model to generalize well to high-resolution images, capturing finer details that are typically hard to interpolate. The experiments showcase the model's robustness and versatility across different types of video data.

**Audience:**

Yes

**Claims And Evidence:**

Yes

**Requested Changes:**

Please supplement the complete multi-resolution experiments, testing on multiple videos and recording the PSNR, SSIM, and LPIPS metrics, since this paper claims that this method is resolution-independent. Improve the description of the method, highlighting the novel contributions proposed by the authors.

**Strengths And Weaknesses:**

The architecture of AdaFNIO ensures that the model is resolution-independent. This allows the model to perform well across various resolutions without the need for retraining on each specific resolution. The model's quasi-linear complexity and efficient design make it suitable for large-scale applications, including those requiring edge computing. This efficiency is crucial for practical implementations in resource-constrained environments .


The experiments with multiple resolutions show that the differences are not very significant, and only the SSIM metric is reported. Furthermore, testing was done on only one scene data. It is expected to see the comprehensive performance across all scenes.
The method proposed in the paper is based on AdaCoF, with only a small part of it being changed. The method section spends a significant amount of space introducing the AdaCoF architecture, but the modifications made by the authors are not described in sufficient detail.

---

### Review · Reviewer_ij39 · 2024-06-25

**Summary Of Contributions:**

This manuscript proposes a physics-informed modification to AdaCoF called AdaFNIO. It evaluates the extended AdaCoF on 4 datasets based on PSNR and SSIM and compares performance to a wide range of baselines. It additionally evaluates the extended AdaCoF with respect to resolution invariance. As a part of ablation studies, the manuscript compares the proposed physics-informed part to other possible physics-informed variants. As a part of limited evaluation on a collection of in-the-wild videos from a broad range domains, the manuscript shows 3 interpolated frames predicted by the extended AdaCoF and using "simple" interpolation.

**Audience:**

Yes

**Claims And Evidence:**

No

**Requested Changes:**

Need to motivate/discuss compromises between on-device/offline processing, compute and memory requirements, and any other more broad considerations in section 1. Take a rover as an example, what could be done here using your technique that cannot be done currently?

Lack of motivating studying micro facial expressions consistently in the manuscript makes the choice of the corresponding dataset rather peculiar in section 3.5.1.

Remove duplicate discussion of compatibility with commonly used examples in section 1.

Give an example of architecture that can generalise from low to high resolution inputs in section 1 page 2.

Explain "generalized in an arbitrary fashion" on page 3.

Explain "overcomes degrees of freedom limitations" on page 3.

Provide clear evidence of efficiency of AdaFNIO claimed on page 3.

UCS101 or UCF101 on page 4?

Explain "improves the degrees of freedom" on page 5.

Explain and introduce every single symbol in equations 2, 3 and 4.

Reduce repeated description of Vimeo90K on page 9.

Explain why 33.36 is bold in Table 1.

State if 0.01, 0.001 or 0.0001 are significant gains in PSNR and SSIM. State statistical significance for results in Table 1 and elsewhere in this manuscript.

Provide evidence of claimed "no performance degradation" on page 10 for examples given in Figure 7.

On page 11 Figure 8 it seems that 0.25 gain in SSIM is called slightly better. Please clarify "slightly better" in the light of significantly smaller gains in SSIM in Table 1.

Explain the need for Table 3 given the information in Figure 8.

Give conclusions to results in Tables 5 and 6.

Provide more quantitative evaluation for experiments in Section 5. Comment on generalisation to more than 3 examples provided.

Remove A Appendix.

**Strengths And Weaknesses:**

The key strength of this manuscript is the part related to resolution invariance.

There are several weaknesses:
- the size of AdaFNIO and often marginal improvements raise a question of suitability of this approach in a broad setting
- the lack of statistical significance information in tables such as Table 1 raises questions regarding what is a gain and what is not
- the lack of study on the impact of noise is concerning in the light of noise robustness being one of the key criticism of optical flow-based methods
- the lack of information about compute requirements (training/inference cost and time) is concerning in the light of compute/battery life requirements stated to be an important factor (note that the first contribution claims efficiency with no evidence provided)
- single mode interpolation - triplets

---

### Note · Authors · 2024-07-12

I have read and agree with the venue's withdrawal policy on behalf of myself and my co-authors.